# Protective Effects of Sodium Para-Aminosalicylic Acid on Lead and Cadmium Co-Exposure in SH-SY5Y Cells

**DOI:** 10.3390/brainsci13030382

**Published:** 2023-02-22

**Authors:** Jian-Chao Peng, Yue Deng, Han-Xiao Song, Yuan-Yuan Fang, Cui-Liu Gan, Jun-Jie Lin, Jing-Jing Luo, Xiao-Wei Zheng, Michael Aschner, Yue-Ming Jiang

**Affiliations:** 1Department of Toxicology, School of Public Health, Guangxi Medical University, Nanning 530021, China; 2Guangxi Colleges and Universities Key Laboratory of Prevention and Control of Highly Prevalent Diseases, Guangxi Medical University, Nanning 530021, China; 3The People’s Hospital of Kaizhou District, Chongqing 405400, China; 4Department of Molecular Pharmacology, Albert Einstein College of Medicine, Bronx, NY 10461, USA

**Keywords:** lead, cadmium, oxidative stress, co-exposure

## Abstract

Background: Combined exposure to lead and cadmium is common in occupational environments. However, the effects of co-exposure to Pb-Cd on neurotoxicity have not been fully clarified. Sodium para-aminosalicylic acid (PAS-Na) has previously been shown to protect neurons from Pb-induced toxicity. This study aimed to investigate the beneficial effect of PAS-Na against co-exposure to Pb-Cd-induced neurodegeneration in SH-SY5Y cells. Methods: The MTT assay was used to detect the effects of Pb and Cd alone, or in combination, on SH-SY5Y cell survival. The effects of Pb and Cd alone or in combination on oxidative stress were assessed by reactive oxygen species (ROS) level. Nrf2, the master switch for antioxidant responses, was detected by immunofluorescence. Protein expression levels of PI3K, Akt, p-Akt, Nrf2 and HO-1 were determined by Western blot analysis. Results: MTT assay results established that the survival rate of SH-SY5Y cells was not significantly affected by exposure to 1 μmol/L lead, 0.25 μmol/L cadmium, and 1-fold Pb-Cd mixture (1 μmol/L Pb + 0.25 μmol/L Cd), while 10-fold Pb-Cd combined exposure (10 μmol/L Pb + 2.5 μmol/L Cd) significantly reduced the survival rate of SH-SY5Y cells. Combined Pb-Cd exposure significantly increased intracellular ROS levels, and N-Acetyl-L-cysteine (NAC) treatment in the 10 μmol/L Pb + 2.5 μmol/L Cd group significantly decreased ROS expression levels, attenuating the levels of oxidative stress. Protein expression of PI3K and p-Akt significantly decreased in the 10 μmol/L Pb + 2.5 μmol/L Cd group, while the expression of PI3K and p-Akt protein increased after PAS-Na intervention. Immunofluorescence analysis showed that levels of Nrf2 in the nucleus increased in the 10 μmol/L Pb + 2.5 μmol/L Cd group, along with Nrf2 protein levels, suggesting that Nrf2 was translocated from the cytoplasm into the nucleus upon combined Pb-Cd exposure. In addition, HO-1 protein expression level, a downstream gene product of Nrf2, was increased. In response to NAC intervention, HO-1 protein expression levels significantly decreased. PAS-Na had the same intervention effect as NAC. Conclusion: Combined exposure to Pb-Cd induced oxidative stress and cytotoxicity in SH-SY5Y cells. PAS-Na displayed antagonistic effects on neurodegenerative changes induced by combined Pb-Cd exposure; hence, it may afford a novel treatment modality for exposure to these metals.

## 1. Introduction

Lead (Pb) and cadmium (Cd) are among the most toxic metals to humans, and their main target organs are the brain and kidney, with ensuing neurotoxicity and/or nephrotoxicity [1,2]. According to the World Health Organization (WHO), lead exposure caused 540,000 deaths in 2016 [3]. Lead and cadmium are symbiotic minerals, and their compounds are a main occupational hazard. Additionally, Pb and cadmium mixtures are common in the environment [4]. The degradation of these two metals is slow, and they persist in the environment for a long time [5,6]. 

Lead exposure may trigger neurodegenerative diseases. A cumulative increase in Pb exposure can aggravate neurological dysfunction related to Parkinson’s disease (PD), leading to decreased cognitive ability [7]. Occupational exposure to Pb has been linked to a longitudinal decline in cognitive function [8]. Several studies have shown that the developing brain is particularly vulnerable to exposure to Pb [9,10], which may have an adverse effect on the brain development of children. Pb exposure can reduce children’s IQ, academic achievement and executive ability, resulting in cognitive impairment [11]. Moreover, neuropsychological research has revealed that Pb exposure can lead to decreases in intelligence, memory, processing speed, comprehension and reading, visuospatial skills and motor skills [12]. Studies in recent years have shown a higher incidence of intellectual disability in children exposed to low Pb compared with high Pb levels [13]. 

Cd exposure can also damage the blood–brain barrier and enter the central nervous system, induce changes in brain morphology and affect neurotransmitter content and enzyme activity [14]. Long-term exposure to Cd at low concentrations has been shown to cause adverse effects on brain metabolism, reducing the levels of norepinephrine, 5-hydroxytryptamine, acetylcholine and other related factors, and damaging the nervous system of the body [14]. Cd is associated with several neurological diseases and intellectual disability in children and may further lead to memory decline. Compared with able-bodied children, serum cadmium content in intellectually disabled children has been shown to be significantly higher, and IQ was negatively correlated with serum cadmium content [15]. In vivo experiments have shown that cadmium exposure in young rats can significantly impair learning and memory in adulthood. 

Oxidative stress is one of the mechanisms of lead- or cadmium-induced neurodegeneration. Lead exposure can disrupt the oxidative balance, reduce the activity of antioxidant enzymes and increase the levels of ROS. Hydrogen peroxide enzyme (catalase, CAT), glutathione peroxidase (GSH-Px) and superoxide dismutase (SOD), as important and oxidative-stress-related enzymes, are target molecules of lead. Additionally, cadmium interacts with mitochondrial binding sites to induce ROS and oxidative stress, disintegrate the membrane potential, increase the activities of CAT- and SOD-related enzymes and reduce intracellular GSH level [16].

The PI3K/Akt signaling pathway is critical for cell regulation. Akt is an important downstream protein in the PI3K signaling pathway. It is activated under the regulation of PI3K and can inhibit apoptosis through other pathways, secondary to inhibition of NF-κB activity and phosphorylation of Bad and Caspase-9. PI3K/Akt and Nrf2/HO-1 signal transduction pathways are important processes of oxidative stress caused by neurocytopathic diseases induced by lead and cadmium exposure [16,17,18,19]. Nrf2/HO-1 activity depends on the PI3K/AKT pathway [20]. The mechanism of neurotoxicity induced by combined exposure to lead and cadmium is not clear yet. When Pb and Cd act alone, the Akt signaling pathway may be activated, causing damage to nerve cells [21,22]. However, there are few studies on the relationship between PI3K/Akt signaling and neurotoxicity during combined exposure to Pb and Cd. 

SH-SY5Y cells, a neuroblastoma cell line, have often been used as a model to study neurotoxic mechanisms [23,24,25]. Sodium para-aminosalicylic acid (PAS-Na) is a tuberculosis drug with non-specific anti-inflammatory effects. Our previous study showed that PAS-Na intervention restored the ultrastructure of hippocampal axons and dendrites to nearly normal in rats upon subchronic lead exposure [26]. PAS-Na has antagonistic effects on learning and memory impairment and hippocampal amino acid neurotransmitter changes in lead-exposed rats [27]. However, whether PAS-Na has an antagonistic effect on nerve cell injury caused by Pb and Cd combined exposure and whether PAS-Na can inhibit oxidative stress and alleviate cell damage have not been reported. Therefore, we aim to explore the protective effects of PAS-Na on lead and cadmium co-exposure in SH-SY5Y cells.

## 2. Materials and Methods

### 2.1. Chemicals

Nrf2 primary antibody and HO-1 primary antibody were purchased from Abcam (Cambridge, UK). Primary antibodies for PI3K, Akt/P-Akt and beta-actin were purchased from Cell Signal Technology (CST, Boston, MA, USA). DMEM culture and fetal bovine serum (FBS) were supplied by Gibco (New York, NY, USA). DMSO and lead acetate were purchased from Sigma-Aldrich (Shanghai) Trading Co, Ltd. (Shanghai, China). Penicillomycin mixture (100 U/mL), 0.25% trypsin, PBS buffer and reactive oxygen species detection kit were acquired from Solarbio (Beijing, China). Cadmium chloride was purchased from Aladdin Reagent (Shanghai, China) Co., Ltd. Methyl tetrazolium (MTT), N-Acetyl-L-cysteine (NAC) and bicinchoninic acid (BCA) protein quantification kit were obtained from Biyuntian High Tech Co., Ltd. (Nantong, China). Sodium para-aminosalicylic acid (PAS-Na) was purchased from Harbin Pharmaceutical Group holding Co., Ltd. (Harbin, China).

### 2.2. Cell Culture

SH-SY5Y cells were cultured in Dulbecco’s modified Eagle medium (DMEM) (Sigma-Aldrich (Shanghai) Trading Co, Ltd., Shanghai, China) high sugar medium supplemented with 10% fetal bovine serum and penicillin/streptomycin (100 U/mL), and maintained at 37 °C with 5% CO_2_ in a humidified atmosphere. The culture medium was replaced twice a week. The SH-SY5Y cells’ morphology were observed with an inverted microscope (OLYMPUS, Tokyo, Japan).

### 2.3. Experimental Treatments

Differentiated SH-SY5Y cells were randomly divided into eleven groups: control, Pb, Cd, Pb + Cd, NAC intervention, NAC control (20 μM), Rosup positive control, 10 μM Pb + 2.5 μM Cd + 100 μM PAS-Na (L-PAS), 10 μM Pb + 2.5 μM Cd + 200 μM PAS-Na (M-PAS), 10 μM Pb + 2.5 μM Cd + 400 μM PAS-Na groups (H-PAS), and PAS-Na control (400 μM). Cells in the Pb group were cultured in medium containing 0, 0.01, 0.1, 0.25, 0.5, 1, 10, 25 and 50 μmol/L lead acetate for 24 h and 48 h, respectively. Cells in the Cd group were cultured in medium containing 0, 0.01, 0.1, 0.25, 0.5, 1, 10, 25 and 50 μmol/L cadmium chloride for 24 h and 48 h, respectively. Cells in the Pb + Cd group were cultured in medium containing 0, 1 μmol/L lead acetate, 0.25 μmol/L cadmium chloride, 1 μmol/L lead acetate + 0.25 μmol/L cadmium chloride, 5 μmol/L lead acetate + 1.25 μmol/L cadmium chloride, and 10 μmol/L lead acetate + 2.5 μmol/L cadmium chloride for 24 h. Cells in the NAC intervention group were pretreated with 20 μM NAC for 1 h, then treated with 10 μM Pb + 2.5 μM Cd for 24 h. Next, the original medium was removed, and cells in L-PAS, M-PAS and H-PAS groups were cultured in medium containing 100, 200 and 400 μM PAS-Na for 24 h, respectively, while the other groups were treated with fresh and complete medium. The concentrations of Pb, Cd and Pb + Cd treatments were selected based on results derived from the MTT assay.

### 2.4. Cytotoxicity Test with the MTT Assay

SH-SY5Y cells were collected and resuspended for cell count. We seeded SH-SY5Y cells at a concentration of 6 × 10^3^ cells/well into 96-well microplates and then randomly assigned them into different groups in different experiments. Each group was set up with 6 duplicate wells. After 24 h of incubation, they were removed the original medium, 110 μL of MTT solution was added to each well and they were kept in the incubator for 4 h at 37 °C; then, 120 μL DMSO was added to dissolve the crystals, and cell viability was measured by taking absorbance at a wavelength of 490 nm. The formula of the cell survival rate was as follows:Cell viability (%) = (experimental group OD − zero adjustment group OD)/(control group OD − zero adjustment group OD) × 100%(1)

### 2.5. ROS Production Detection 

SH-SY5Y cells were planted into 12-well plates (10 × 10^4^ cells/well) and cultured for 24 h; each group had 3 duplicate wells. The detection was performed strictly in accordance with the manufacturer’s instructions. Briefly, the harvested cells were incubated with diluted DCFH-DA (10 μmol/L final concentration) for 50 min in the dark at 37 °C. After washing with PBS, the images were observed and collected under a fluorescence microscope. Fluorescence was visualized with the EVOS fluorescence microscopy imaging system (Thermo Fisher; Waltham, MA, USA). The ROS levels were expressed as the folds of control.

### 2.6. Immunofluorescence

SH-SY5Y cells were seeded in 24-well plates at the concentration of 3 × 10^4^ cells/well and incubated for 24 h at 37 °C; each group had 3 duplicate wells. After being treated with 0, 10 μmol/L Pb + 2.5 μmol/L Cd for 24 h, cells were fixed with 4% paraformaldehyde (Solarbio, Beijing, China) for 10 min at room temperature, followed by membrane permeabilization using 0.1–0.25% Triton X-100 (Preparation with PBS). Cells were blocked with 1% bovine serum albumin (BSA) (Biyuntian High Tech Co., Ltd., Nantong, China) and incubated with 300 μL of a diluent of primary antibody (Biyuntian High Tech Co., Ltd., Nantong, China) at 4 °C overnight. Next, the cells were incubated with a diluent of secondary antibodies (Biyuntian High Tech Co., Ltd., Nantong, China) for 1 h in the dark. Samples were washed three times with Phosphate Buffer Saline (PBS), stained with 4′,6-diamidino-2-phenylindole (DAPI) (in the dark) for 1 min and visualized and photographed under a fluorescence microscope. 

### 2.7. Western Blotting Assay

Briefly, proteins from SH-SY5Y cells were extracted, then we followed the instructions of the BCA Protein Quantification Kit seriously to determine the concentration of the proteins. Protein extracts were loaded onto the gel for electrophoresis and then transferred to a polyvinylidene fluoride (PVDF) membrane. After blockade with 5% skim milk, membranes were incubated with primary antibodies (Nrf2 [1:1000, Abcam, Cambridge, UK], HO-1 [1:1000, Abcam, Cambridge, UK], PI3K [1:1000, CST, Boston, MA, USA], Akt [1:1000, CST, Boston, MA, USA], Beta-actin [1:1000, CST, Boston, MA, USA]) overnight at 4 °C. Then, we transferred the PVDF membrane to secondary antibodies and incubated them at room temperature for 1 h. Finally, protein bands were visualized by using an enhanced chemiluminescence system (Thermo Fisher, Waltham, MA, USA) and qualified using Image J.

### 2.8. Statistical Analysis

Statistical analyses were conducted using SPSS 22.0 software (IBM, Somers, NY, USA), and the results were expressed as mean ± standard deviation (SD) of at least three independent experiments. Multiple groups of samples were compared using one-way analysis of variance (ANOVA) and pairwise comparison with the LSD test. With α = 0.05 as the test level, *p* < 0.05 was considered statistically significant.

## 3. Results

### 3.1. Establishment of SH-SY5Y Cell Nerve Injury Model Induced by Pb and Cd 

To ensure the success of the cellular nerve damage model, first, cell survival was assessed in SH-SY5Y cells upon Pb or Cd treatment. SH-SY5Y cells exposed to the highest concentration of 1 μmol/L Pb and 0.25 μmol/L Cd for 24 and 48 h showed no significant effect on growth compared with controls (*p* > 0.05, Figure 1A,B). Therefore, the exposure time was set to 24 h, Pb exposure was set to 1μmol/L and the Cd exposure was set to 0.25 μmol/L. Compared with the control group, the viability of SH-SY5Y cells in 1 μmol/L Pb, 0.25 μmol/L Cd and 1 μmol/L Pb + 0.25 μmol/L Cd groups were not significantly affected and were statistically indistinguishable. The survival rate of SH-SY5Y cells in 10 μmol/L Pb + 2.5 μmol/L Cd group was significantly decreased (*p* < 0.05, Figure 1C). Microscopic observation of cell morphology showed that compared with the control group, the number of normal cells decreased while the dead cells increased after 10 μmol/L Pb + 2.5 μmol/L Cd treatment. In addition, the cells became swollen, and intercellular connections and cellular eminence were reduced. (Figure 1D).

### 3.2. Co-Exposure of Lead and Cadmium Can Induce Oxidative Stress in SH-SY5Y Cells 

ROS expression level in 1 μmol/L Pb + 0.25 μmol/L Cd group and 10 μmol/L Pb + 2.5 μmol/L Cd group was significantly higher than that in the control group (*p* < 0.05, Figure 2A–D). Compared with 10 μmol/L Pb + 2.5 μmol/L Cd group, ROS expression level in the NAC intervention group was significantly decreased (*p* < 0.05, Figure 2C,D), which indicates that oxidative stress was inhibited. Compared with the control group, there was no significant difference in ROS expression level in the NAC control group (*p* > 0.05, Figure 2C,D).

### 3.3. Effects of Single and Combined Exposure of Lead and Cadmium and Sodium para-aminosalicylic acid (PAS-Na) Intervention on PI3K/Akt Signaling Pathway

Compared with the control group, expression of PI3K protein in SH-SY5Y cells upon 10 μmol/L Pb + 2.5 μmol/L Cd exposure was significantly decreased, while the expression of Akt protein was not significantly altered (*p* < 0.05, Figure 3A,B,D,E), and the p-Akt/Akt protein ratio was decreased (*p* < 0.05, Figure 3C,F). Thus, indicating that combined exposure to Pb and Cd activated the PI3K/Akt pathway (*p* < 0.05, Figure 3D), and after 24 h of PAS-Na intervention, compared with 10 μmol/L Pb + 2.5 μmol/L Cd group, the expression of PI3K significantly increased at all three PAS-Na concentrations (*p* < 0.05, Figure 3D,E). The expression of p-Akt was significantly increased in SH-SY5Y cells after 400 μM PAS-Na treatment (*p* < 0.05, Figure 3D,F). Compared with the control group, there was no significant difference in p-Akt protein expression in SH-SY5Y cells in the PAS-Na control group (*p* > 0.05, Figure 3D,F).

### 3.4. PAS-Na Attenuates the Increase in Nrf2 and HO-1 Protein Expression Caused by Lead and Cadmium Co-Exposure

The distribution of Nrf2 in SH-SY5Y cells after combined Pb and Cd exposure was detected by immunofluorescence. Compared with the control group, Nrf2 increased in the 10 μmol/L Pb + 2.5 μmol/L Cd group (*p* < 0.05, Figure 4A,B), suggesting that Nrf2 translocated from the cytoplasm into the nucleus upon combined Pb and Cd treatment. Nrf2 and HO-1 protein expression was detected by Western blot analysis. The results showed that compared with the control group, protein expressions of Nrf2 and HO-1 significantly increased after treatment with 10 μmol/L Pb + 2.5 μmol/L Cd for 24 h (*p* < 0.05, Figure 4B–G). After 1 h NAC intervention, HO-1 protein expression was significantly decreased compared with the 10 μmol/L Pb + 2.5 μmol/L Cd group (*p* < 0.05, Figure 4F–H). Compared with the control group, there was no significant difference in Nrf2 and HO-1 protein expression in SH-SY5Y cells in the NAC control group (*p* > 0.05, Figure 4F–H). Compared with 10 μmol/L Pb + 2.5 μmol/L Cd group, 200 μM PAS-Na treatment decreased the expression of HO-1 protein in SH-SY5Y cells after 24 h treatment (*p* < 0.05, Figure 4I,K), while Nrf2 expression was decreased after 400 μM PAS-Na intervention(*p* < 0.05, Figure 4I,J). Compared with the control group, there was no significant difference in Nrf2 and HO-1 protein expression in SH-SY5Y cells in the PAS-Na control group (*p* > 0.05, Figure 4I–K).

## 4. Discussion

Environmental pollution and exposure to heavy metals (such as arsenic, lead, cadmium and mercury) continues to pose a serious global health risk [28]. Early exposure to Pb and Cd can heighten children’s risk for aberrant neurodevelopment, resulting in lifelong physical, intellectual and behavioral disorders [29]. Pb and Cd are non-metabolizable, with continuous exposure leading to metal accumulation [30]. In vitro studies have previously established that neurite outgrowth in PC12 cells was inhibited by combined exposure to Pb and Cd [31]. The decrease in the survival rate upon combined exposure to Pb-Cd PC12 cells was more pronounced compared with Pb exposure alone, and the inhibition of the outgrowth of neurites was enhanced [32]. Herein, we found that, alone, Pb and Cd exposures were toxic to SH-SY5Y cells in a concentration-dependent manner. Cells treated with combined Pb and Cd (Pb 10 μmol/L + Cd 2.5 μmol/L) showed as morphologically swollen with synaptic fracturing. However, when Pb and Cd were combined at the maximal non-effective concentration (Pb 1.0 μmol/L + Cd 0.25 μmol/L), there was no significant difference in the survival rate compared with cells exposed to a single metal, likely related to the short exposure time and low concentrations inherent to this study. 

Oxidative stress is a key mechanism of toxicity of both Pb and Cd [33,34,35,36], secondary to the binding of oxygen, nitrogen and sulfur ligands, affecting the expression of various enzymes and proteins [33,37]. Cd promotes the production of ROS and lipid peroxidation (LPO) in the brain. Due to its high oxygen consumption and low antioxidant defense activity, the central nervous system (CNS) is most sensitive and highly susceptible to oxidative stress. In vivo studies have shown that combined Pb, Cd and Hg low-dose exposure induced oxidative stress, altered intracellular free calcium and increased apoptosis in the CNS of rats [38]. Exposure to a mixture of Pb, Cd and Hg caused a dose-dependent impairment in learning, memory and perception in rats, and combined metal exposure disrupted synaptic remodeling, which is in dendritic spine growth, maintenance and elimination [39]. Epidemiological investigations have also shown that higher urinary As, Cd and Pb levels are associated with increased markers of oxidative stress [40]. Ayat et al. have shown that Pb and Cd can affect bioenergetics in human osteoblasts, induce redox stress and limit the effectiveness of the cellular antioxidant defense system by increased production of ROS [41]. Here, ROS expression was increased significantly after combined Pb and Cd exposure. NAC is a commonly used antioxidant in oxidative stress experiments, which can effectively protect the body from oxidative stress damage caused by other toxic substances. Our results demonstrate that ROS levels were significantly decreased in cells treated with NAC, Pb and Cd, and oxidative stress was significantly inhibited, compared with the combined exposure of Pb and Cd. Collectively, these studies establish that NAC afforded protection from the cytotoxicity of Pb and Cd.

Nrf-2 is a major player in the endogenous antioxidant system, plays a key role in ROS and upregulates other endogenous genes, such as hemoxygenase-1 (HO-1), neutralizes ROS and LPO and protects against oxidative stress-induced neurodegeneration [42,43]. Nrf2 translocates into the nucleus after combined Pb and Cd exposure and activates downstream HO-1 gene expression, affording a protective effect against oxidative stress [41]. Using immunofluorescence detection, we found that Nrf2 expression increased in the nucleus of SH-SY5Y cells exposed to Pb and Cd. Western blotting showed that combined Pb and Cd exposure significantly increased the expression of Nrf2 downstream protein HO-1, and combined exposure to NAC and Pb and Cd attenuated the adverse effects of Pb and Cd. Accordingly, this protective effect seems to be mediated by activating the Nrf2-Keap1 antioxidant pathway. After the intervention of Sodium para-aminosalicylic acid (PAS-Na) on SH-SY5Y cells combined exposed to lead and cadmium, the protein expression of Nrf2 and HO-1 was decreased. The antioxidant effect of PAS-Na was consistent with that of NAC, and both agents reduced the cytotoxic effect caused by lead and cadmium combined exposure. These results showed that PAS-Na can inhibit the activation of the Nrf2 pathway induced by the combined action of Pb and Cd by blocking oxidative stress.

The PI3K/Akt pathway plays an important role in the toxicity of exposure to many heavy metals [44,45]. Pb exposure can inhibit the PI3K/Akt pathway, and induce apoptosis in chicken lymphocytes in vitro [21], while Cd exposure has been shown to induce apoptosis through the PI3K-Akt-mTOR signaling pathway in vitro [46]. Li et al. have showed that Pb^2+^-induced hippocampal neuronal death was mediated through PI3K/AKT pathway activation [47]. A previous study has established that p-AKT can activate the Nrf2/HO-1 signaling pathway [48]. Interestingly, in this study, PI3K and p-Akt expressions decreased in response to combined exposure to Pb and Cd, which was not inconsistent with the increase on Nrf2 and HO-1 proteins, which may be related to the time and dose of combined exposure to lead and cadmium, indicating the PI3K/AKT pathway had been restrained. However, in the present study, after PAS-Na’ s intervention, the expression of PI3K and p-Akt increased, indicating that PAS-Na can have an antagonistic effect on the reduction of PI3K and p-Akt. Herein, it is reasonable to speculate that PAS-Na can exert this protective effect through the PI3K/p-Akt pathway.

PAS-Na is a non-specific anti-inflammatory drug for tuberculosis. Our group found in previous experiments that PAS-Na intervention could increase the activity of antioxidant enzymes in primary basal ganglia or hippocampal neurons exposed to Mn [49,50]. PAS-Na attenuates Mn-induced NLRP3 inflammasome-dependent apoptosis by inhibiting NF-κB pathway activation and oxidative stress [51]. Additionally, we found that PAS-Na has a certain antagonistic effect on PC12 cell apoptosis induced by Pb, and the mechanism may be related to PAS-Na’s anti-lipid peroxidation damage and the reduction of Bax/Bcl-2 ratio [52]. Herein, we showed that PAS-Na’s intervention in SH-SY5Y cells after combined Pb and Cd treatment reduced Nrf2 and HO-1 protein expression, and PAS-Na treatment was congruent with the effect of NAC, attenuating the Pb-Cd combined effect, suggesting that PAS-Na can inhibit the activation of the Nrf2 pathway induced by the combined action of Pb-Cd by blocking oxidative stress. CaNa2EDTA is known to be commonly used for lead discharge, but it does not penetrate the blood–brain barrier for stimulatory effect. PAS-Na and its major metabolite N-acetyl-para-aminosalicylic acid (AcPAS) can cross the blood–brain barrier into brain, which exerts therapeutic effect on Mn exposure in rats [53,54]. PAS-Na treatment promoted Mn excretions and recovered the changes in Glu turnover induced by Mn [55], and also decreased Mn levels in the whole blood and selected brain tissue [56]. Moreover, PAS-Na has low molecular weight, in part assuring its rapid access to the site of action [57], thus directly promoting brain Pb and Cd chelation. In addition, PAS-Na is a non-steroidal anti-inflammatory drug. Our previous studies showed that PAS-Na reduced Pb-induced ultrastructural damage in the hippocampus and afforded neuroprotection. The IP3R-Ca^2+^-ASK1-p38 signaling pathway has also been shown to mediate Pb-induced apoptosis in hippocampal neurons, and PAS-Na ameliorated this effect [58]. Besides, we found that PAS-Na may antagonize the neurotoxic effects of lead exposure through the expression of the SIRT1/HMGB1/NF-κB pathway and BDNF [59]. Taken together, our novel results illustrate that PAS-Na can attenuate the neurotoxicity induced by combined Pb and Cd treatment through its anti-inflammatory properties, providing reference values for future studies. Additional studies should be directed at the therapeutic efficacy of PAS-Na in attenuating in vivo combined Pb-Cd treatment-induced neurological impairment, to ascertain its efficacy in attenuating neurotoxicity and identifying putative targets and pharmacological modalities for its action. 

## 5. Conclusions

In conclusion, our novel study established that the combined exposure to Pb-Cd induced oxidative stress in SH-SY5Y cells, in turn reducing the expression of PI3K and p-Akt proteins and increasing the expression of Nrf2 and HO-1 proteins, consistent with increased ROS levels. PAS-Na attenuated the increased expression of Nrf2 and HO-1 proteins caused by the combined metal exposure, and concomitantly increased the expression of PI3K and p-Akt proteins. These findings provide insight into the therapeutic mechanisms of PAS-Na on Pb-Cd induced neurotoxicity. In addition, further studies are needed to elucidate the therapeutic mechanism of PAS-Na on combined exposure to Pb and Cd.

## Figures and Tables

**Figure 1 brainsci-13-00382-f001:**
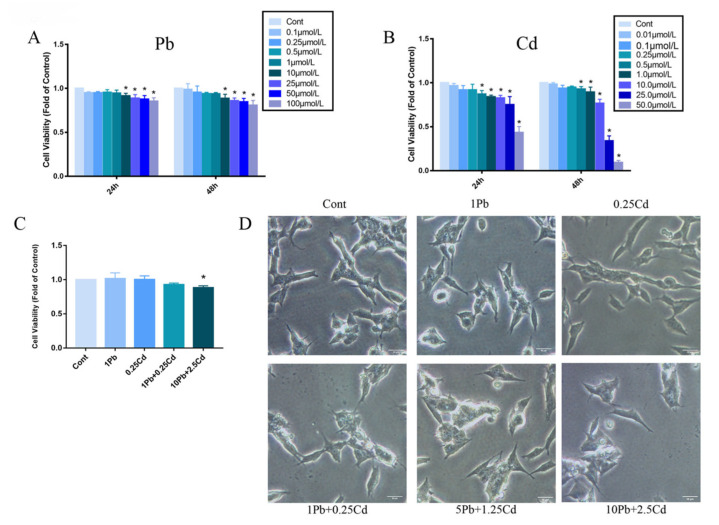
Survival rate of SH-SY5Y cells under different concentrations of Pb and Cd for 24 and 48 h. (**A**,**B**) SH-SY5Y cells were treated with Pb (0~100 μmol/L) or Cd (0~50 μmol/L) for 24 and 48 h. (**C**) Survival rate of SH-SY5Y cells under lead and cadmium co-exposure for 24 h. An amount of 1 Pb is 1 μmol/L Pb, and 0.25 Cd is 0.25 μmol/L Cd. An amount of 1 Pb + 0.25 Cd is the mixture of 1 μmol/L Pb + 0.25 μmol/L Cd; 10 Pb + 2.5 Cd is the mixture of 10 μmol/L Pb + 2.5 μmol/L Cd. (**D**) Morphological changes of SH-SY5Y cells were observed after co-exposure to Pb and Cd for 24 h. Scale bars: 50 μm. *, *p* < 0.05 compared with the control, data are presented as mean ± SD.

**Figure 2 brainsci-13-00382-f002:**
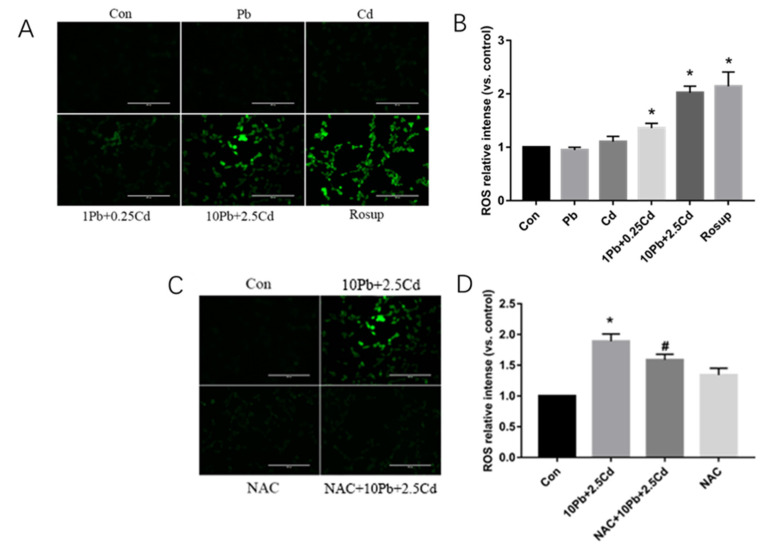
Effects of lead and cadmium alone or co-exposure on ROS levels in SH-SY5Y cells. Cells in the positive control group were treated with Rosup for 1 h after culturing with normal medium for 23 h. The level of intracellular ROS production was measured using a DCFH-DA fluorescent probe (**A**,**C**), and the fluorescence intensity was analyzed by Image J (**B**,**D**). An amount of 1 Pb is 1 μmol/L Pb, and 0.25 Cd is 0.25 μmol/L Cd. An amount of 1 Pb + 0.25 Cd is 1 μmol/L Pb + 0.25 μmol/L Cd mixture; 10 Pb + 2.5 Cd is 10 μmol/L Pb + 2.5 μmol/L Cd mixture. *, *p* < 0.05 compared with the control. #, *p* < 0.05, compared with the 10 Pb + 2.5 Cd. Data are presented as mean ± SD. Scale bar = 200 μm.

**Figure 3 brainsci-13-00382-f003:**
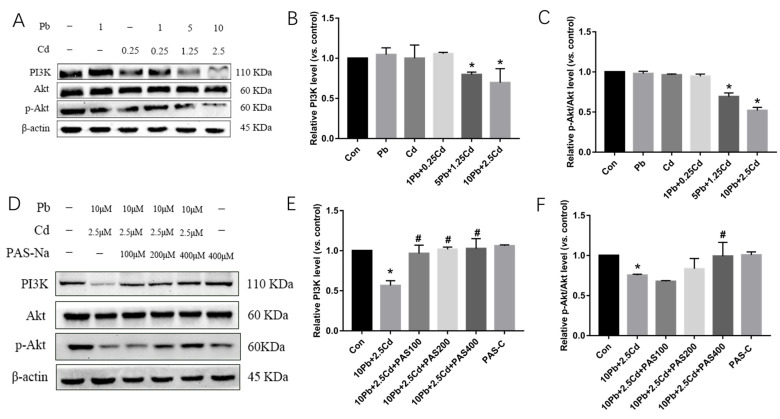
Effects of lead and cadmium on PI3K/Akt in SH-SY5Y cells, and effects of PAS-Na on the expression of PI3K and Akt exposed to lead and cadmium. (**A**,**D**) Western blot results; (**B**,**E**) PI3K expression; (**C**,**F**) p-Akt/Akt ratio. An amount of 1 Pb is 1 μmol/L Pb, and 0.25 Cd is 0.25 μmol/L Cd. An amount of 1 Pb + 0.25 Cd is 1 μmol/L Pb + 0.25 μmol/L Cd mixture, 5 Pb + 1.25 Cd is 5 μmol/L Pb + 1.25 μmol/L Cd mixture, 10 Pb + 2.5 Cd is 10 μmol/L Pb + 2.5 μmol/L Cd mixture. *, *p* < 0.05 compared with the control. #, *p* < 0.05, compared with the 10 Pb + 2.5 Cd. Data are presented as mean ± SD.

**Figure 4 brainsci-13-00382-f004:**
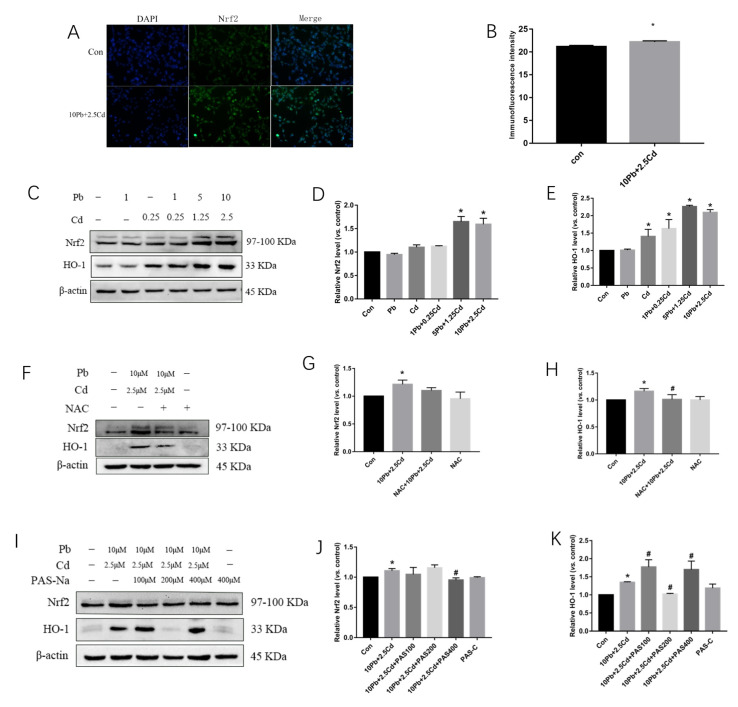
Effects of lead and cadmium exposure alone or in combination on Nrf2/HO-1 in SH-SY5Y cells. Additionally, the effect of NAC and PAS-Na intervention on Nrf2 and HO-1 protein expression exposed to lead and cadmium. (**A**) Immunofluorescence staining showed the location of Nrf2 protein in basal ganglia after combined Pb-Cd exposure at 24 h. Scale bar = 20 μm. (**B**) Immunofluorescence intensity in the control group and 10 Pb + 2.5 Cd group. (**C**,**F**,**I**) Western blot results; (**D**,**G**,**J**) Nrf2 expression; (**E**,**H**,**K**) HO-1 expression. An amount of 1 Pb is 1 μmol/L Pb, and 0.25 Cd is 0.25 μmol/L Cd. An amount of 1 Pb + 0.25 Cd is 1 μmol/L Pb + 0.25 μmol/L Cd mixture, 5 Pb + 1.25 Cd is 5 μmol/L Pb + 1.25 μmol/L Cd mixture, 10 Pb + 2.5 Cd is 10 μmol/L Pb + 2.5 μmol/L Cd mixture. *, *p* < 0.05 compared with the control. #, *p* < 0.05, compared with the 10 Pb + 2.5 Cd. Data are presented as mean ± SD.

## Data Availability

All data generated or analyzed during this study are included in this published article.

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
