# Peer review of "Protective Effects of Sodium Para-Aminosalicylic Acid on Lead and Cadmium Co-Exposure in SH-SY5Y Cells"

_brainsci, 2023, doi:10.3390/brainsci13030382_

Round 1

Reviewer 1 Report

Comments and Suggestions for Authors

This manuscript investigates the effect of PAS-NA on Lead and Cadmium exposure in a neuroblastoma cell line. The authors showed a protective effect of the compound on metal exposure, the experiments’ results are very interesting but the results part needs to be carefully reviewed.

Several comments need to be addressed:

1.    SHSY-5Y cells, a neuroblastoma cell line, are used as a model to nerve injury. Can the author explain what do they mean and cite any references where this cell line has been used as nerve injury model?

2.    In section 3.1, line 169. What do the authors mean with the highest concentration? Also, if values were set at 1μmol/L for Pb and 0.25μmol/L for Cd, why the next experiments are conducted using higher concentrations?

3.    In section 3.1 and figure 1, different concentrations of Pb and Cd were used. Can the authors provide a rationale of why they used these concentrations? Also, since toxicity has been showed starting from 1μmol/L for Pb and 0.25μmol/L for Cd, why not test concentrations between 1 and 10 umol/L such as 2.5 umol/L and 5 umol/L?

4.    Figure 1, panel D: can the authors provide higher magnification? It is hard to appreciate differences in morphology. In this panel, the authors show data using concentration of 5 umol/L for Pb and 1.25 umol/L for Cd. Why were these concentrations not used in panel A and B? Also, the scale bar is missing.

5.    Section 3.2: Very hard to read the explanation of the results. Some words are repeated. What do the authors mean with control group? Which one is the control group? Please explain why NAC is used in this experiment.

6.    Figure 2, panel C and D. I can not see any green cells in NAC and NAC+10Pb+2.5Cd, could the author explain how the quantification was performed or provide some other pictures? Also, why were 1Pb+0.25Cd concentrations not used with NAC?

7.    Section 3.3: The part “Compared with the control group, expression of PI3K protein in SH-SY5Y cells upon 203 10μmol/L Pb+ 2.5μmol/L Cd exposure was significantly decreased” seems to be repeated. Not clear what the authors mean with control group.

8.    It is not clear why in some experiments the concentration of 5 umol/L for Pb and 1.25 umol/L for Cd were used, and in other experiments they were not used. Can the authors elaborate more on this?

9.    In the discussion part, the authors stated that: “there was no significant difference in the survival rate compared to cells exposed to a single metal, likely related to the short exposure time and low concentrations inherent to this study.” Can the authors elaborate on the reasons they did not repeat the experiments with higher concentrations and longer exposure? It is not clear to me why they have chosen Pb 1.0μmol/L and Cd 0.25μmol/L in some experiments, Pb 5 and Cd 1.25 in others and Pb 10 and Cd 2.5 in others.

10.  Reference 25. Please double check the reference.

Minor revisions:

1.    Lane 83: Caspase is spelled wrong

2.    Section 2.4. The term “Non-infected” seems to be wrong.

3.    Figure 1, panel C: please provide concentration of Pb and Cd where they were exposed alone.

4.    Figure 2, panel A, B: please provide concentration of Pb and Cd where they were exposed alone.

5.    Figure 3, panel B,C: please provide concentration of Pb and Cd where they were exposed alone.

6.    Figure 3, panel D: hard to read the concentration used. Is it possible to make the panel bigger?

7.    Figure 4, panel A: A quantification graph could be useful to complete the figure.

8.    Figure 4, panel C,D: please provide concentration of Pb and Cd where they were exposed alone. Figure 4, panel H: same comment as for Fig 3, panel D. Hard to read concentrations.

Reviewer 2 Report

Comments and Suggestions for Authors

This is a logical and well laid out manuscript detailing the ability of an antioxidant small molecule to alleviate the toxicity caused by mixtures of Pb and cd. The conclusions are overall supported by the data. Minor comments: 

In the abstract, the following sentence is a fragment: "While 10-fold Pb-Cd combined exposure (10μmol/L Pb+2.5μmol/L 25 Cd) significantly reduced the survival rate of SH-SY5Y cells."

In the figure 2 legend, ROSUP should be explained. Is this a positive control?

Our previous study showed that PAS-Na intervention restored the ultrastructure of hippocampal axons and dendrites to nearly normal" is not clear - under what conditions? 

The superscripts need to be fixed for references and scientific notation throughout. 

It would be helpful to include the chemical structure of PAS-Na.

Round 2

Reviewer 1 Report

Comments and Suggestions for Authors

The authors have addressed all the questions and comments.

Author Response

Thank you.